# Polyethyleneimine-Based Drug Delivery Systems for Cancer Theranostics

**DOI:** 10.3390/jfb14010012

**Published:** 2022-12-23

**Authors:** Chong Zhao, Benqing Zhou

**Affiliations:** 1Department of Biomedical Engineering, College of Engineering, Shantou University, Shantou 515063, China; 2Guangdong Provincial Key Laboratory for Diagnosis and Treatment of Breast Cancer, Shantou 515041, China

**Keywords:** polyethylenimine, drug delivery, cancer treatment, cancer imaging, cancer theranostics

## Abstract

With the development of nanotechnology, various types of polymer-based drug delivery systems have been designed for biomedical applications. Polymer-based drug delivery systems with desirable biocompatibility can be efficiently delivered to tumor sites with passive or targeted effects and combined with other therapeutic and imaging agents for cancer theranostics. As an effective vehicle for drug and gene delivery, polyethyleneimine (PEI) has been extensively studied due to its rich surface amines and excellent water solubility. In this work, we summarize the surface modifications of PEI to enhance biocompatibility and functionalization. Additionally, the synthesis of PEI-based nanoparticles is discussed. We further review the applications of PEI-based drug delivery systems in cancer treatment, cancer imaging, and cancer theranostics. Finally, we thoroughly consider the outlook and challenges relating to PEI-based drug delivery systems.

## 1. Introduction

Cancer is a human disease characterized by abnormal cell proliferation and metastasis and poses a very significant threat to human health. Its occurrence is closely related to harmful environment, bad lifestyle, and heredity. Early diagnosis and treatment of cancer is the most important strategy to improve survival rates. Recently, nanotechnology has attracted extensive attention in the biomedical field, particularly in the early diagnosis and treatment of cancer [1,2]. 

The development of novel multifunctional nanoparticles (NPs) for cancer theranostics is one of the most important trends in the development of nanomedicine [3,4,5,6]. Compared with traditional drug delivery systems, NP-based drug delivery systems can not only improve the water solubility and stability of drugs, but also influence the distribution of drugs in vivo owing to the nanosized effects of NPs [7]. In addition, the kinetics of drug release can be controlled through material design and surface modifications. More importantly, targeted molecules can be modified on the surface of NPs to specifically target tumor sites, thereby improving the bioavailability of drugs and reducing toxicity to off-target tissues [8]. To date, various NPs have been developed for construction of NP-based drug delivery systems including liposomes [9,10], micelles [11], nanogels [12,13], radionuclide-labeled NPs [14,15], and metal NPs [16,17,18,19]. 

Polyethyleneimine (PEI) is a cationic polymer molecule composed of abundant amine groups and two aliphatic carbons, and because of its specific structure and properties it has been widely used to stabilize or modify various inorganic hybrid NPs [20]. As a cationic polyamine, PEI can interact with or bind to anionic residues of DNA templates and polymerase through electrostatic interaction, thus significantly improving their transfection efficiency [21]. In addition, the strong positive surface potential of PEI presents obvious cytotoxicity to cells because of its abundant amine groups [21]. Therefore, neutralizing the surface potential of PEI through various chemical or physical modifications can effectively reduce its cytotoxicity and improve biocompatibility. It is worth noting that these surface modifications not only improve the biocompatibility of PEI, but also enable it to acquire other functions, such as biomarker and targeting.

Currently, there are very few recent and systematic reviews concerning PEI-based drug delivery systems for cancer theranostics. In this paper, we first summarize PEI modifications, including biocompatibility and functional modifications. Secondly, the synthesis of PEI-based NPs for biomedical applications is introduced. Thirdly, we comprehensively review the applications of PEI-based drug delivery systems in cancer treatment, imaging, and theranostics. Finally, the current challenges and further prospects related to PEI-based drug delivery systems are discussed.

## 2. Overview of PEI

PEI is a commercially widely used cationic polymer containing primary, secondary, and tertiary amino groups in a ratio of 1:2:1 with strong positive charges [22]. PEI can be synthesized as linear PEI (Figure 1a) or branched PEI (Figure 1b) with a molecular weight ranging from 700 Da to 1000 kDa according to the degree of polymerization [23]. PEI can be easily prepared using an AB-type monomer via a simple one-step reaction [24]. In addition, PEI can be considered a low-cost option compared to dendrimers with the same molecular weight [25]. PEI has been widely used in different fields because of its unique structure and abundant amino groups. For example, in industry, PEI can be used as a flocculant to remove oil present in synthetically produced water, or as a wet strength agent in paper-making and the manufacture of shampoo [26,27]. In biomedicine, PEI is widely used in enzyme immobilization [28], virus immobilization on cellulose [29], cell adhesion [30], gene transfection [31], and the synthesis of NPs to enhance their stability and anticancer efficacy [32].

Branched PEI is a hyperbranched polymer synthesized using the monomer method; that is, the cationic polymer is obtained by acid-catalyzed ring-opening polymerization of aziridine monomers [33]. Each branch of secondary amines in the branched chain of hyperbranched PEI has 3–35 nitrogen atoms on average. This branch distribution can form a spherical internal structure, which can encapsulate NPs, drug molecules, and other small molecules. Furthermore, the lone pair electrons of nitrogen atoms in branched PEI can stabilize metal ions via coordination interaction. Therefore, branched PEI has a wide range of applications in gene transfection [34,35,36], drug delivery [37,38], and molecular imaging [25,39].

Linear PEI contains only secondary amines, whereas branched PEI contains various types of amines, i.e., primary, secondary, and tertiary. Linear PEI is solid at room temperature, in contrast to branched PEI which is liquid at all molecular weights [20]. Linear PEI is a high-charge cationic polymer and has been widely used in biomedical fields. For example, linear PEI has antibacterial properties against various pathogens and can therefore be used as a bacteriostatic agent [40,41]. Additionally, as a cationic polymer, linear PEI can form a polymer with nucleotides for gene transfer [42]. Compared with branched PEI, linear PEI is an effective nonviral gene vector with higher cell viability and transfection efficiency [43]. We have presented a balanced picture of the PEI studies including advantages and limits in Figure 2.

## 3. PEI Modifications

As a cationic polymer, PEI contains abundant amino groups and as a result has a certain degree of cytotoxicity. Cationic PEI enters cells by adhering to negatively charged transmembrane heparanproteoglycans, which can cause cell damage through membrane destabilization [44]. Additionally, the internalized PEI causes apoptotic cell death by forming pores in the mitochondrial membrane [45,46]. PEI is not well-degraded in organisms, and its cytotoxicity is closely related to its molecular weight and branching degree [47]. Branched PEI with a higher molecular weight has a higher cytotoxicity. The surface amines of PEI can be shielded with simple modifications, thus significantly improving the biocompatibility of PEI [21]. At present, the surface amines of PEI are mainly shielded with covalent bonds such as carboxylation, acetylation, and hydroxylation, or with electrostatic modification of negatively charged proteins. However, currently, there is a lack of systematic research to contrast the benefits and challenges of these approaches for the surface modifications of PEI. For example, Wen et al. improved the biocompatibility of PEI through carboxylation, acetylation, hydroxylation, and PEGylation [21]. These methods effectively reduced or shielded the positive charge of the PEI, thus reducing cytotoxicity. As shown in Figure 3, various functional groups including polyethylene glycol (PEG), folic acid (FA), hyaluronic acid (HA), fluorescent tags, and protein can be modified with PEI for biomedical applications [24,25,48,49,50,51,52,53,54,55,56,57,58]. We summarize PEI modifications for biomedical applications in recent years in Table 1.

### 3.1. Carboxylation Modification

PEI is a representative nonviral vector for gene delivery because the proton sponge of PEI can protect nucleic acids from nuclease digestion [79,80]. Carboxylation can effectively improve the biocompatibility of PEI, and carboxylated PEI is often used for gene delivery. For example, Nakamura et al. constructed a gene vector by forming an amide bond between the carboxyl group of a peptide and the primary amino group of PEI [60]. The carboxyl-modified PEI was used as a gene carrier in response to tumor-specific activation of protein kinase C alpha (PKCα) to release plasmid DNA (pDNA) for gene expression. In another study, Nam et al. conjugated polylactic-co-glycolic acid (PLGA) with a carboxyl terminal group to PEI to synthesize novel amphiphilic block copolymers self-assembled in water [81]. This work indicated that PEI–PLGA aggregates were easily adsorbed to the cell surface and transferred to the cytoplasm, and thus could be used as effective drug carriers.

### 3.2. Acetylation Modification

Similarly, acetylation modification can also reduce the cytotoxicity of PEI and improve the efficiency of gene delivery [63,64]. Calarco et al. found that acetylation of PEI significantly reduced the cyto- and genotoxicity of PEI-based NPs [65]. The acetylated PEI-based NPs promoted DNA intake and reduced the production of reactive oxygen species (ROS) responsible for DNA damage. In our previous work, we found that acetylation modification effectively reduced the positive charges of PEI and thus improved its biocompatibility for cancer imaging and therapy applications [24,54,55,56,57,58].

### 3.3. Hydroxylation Modification

The introduction of hydroxyl groups on the PEI surface is often used in the fields of gene and drug delivery. Wu et al. synthesized biodegradable chitosan-g-PEI-g-PEG-OH copolymer for gene transfection [66]. The PEI-grafted chitosan significantly reduced the toxicity of the PEI and had no effect on gene transfection efficiency. Hydroxyl modification on the PEI surface can improve its tolerance to serum and reduce the cytotoxicity of the nucleic acid carrier. Notably, the hydroxyl group could act as a bridge to link PEI to other functional groups such as FA, antibodies, and other targeted molecules. Chen et al. reported that hydroxyl-modified PEI showed lower cytotoxicity and better serum-resistant capacity than free PEI for the delivery of nucleic acids [67]. In HeLa cells containing serum, the transfection efficiency of hydroxy-modified PEI was 29 times higher than that of free PEI. Furthermore, the hydroxyl-modified PEI/siRNA complexes displayed a stronger knockdown effect in CT26 cells.

### 3.4. PEG Modification

PEG is widely used in drug delivery systems because of its high water solubility, nonimmunogenicity, and excellent biocompatibility [82]. Different types of PEGs with various chain lengths have been conjugated with PEI to improve the stability and transfection efficiency of PEI [69]. Studies have shown that the degree of PEGylation and the molecular weight of PEG have a significant influence on the properties of PEI [24]. The stability and transfection efficiency of PEI/DNA complexes were affected by graft length and the PEG side chains [69,70]. PEG side chains stabilized PEI/DNA complexes in the presence of salt; however, intracellular gene delivery was also interrupted by longer PEG side chains because of their more effective spatial obstruction [70]. Short PEG side chains with a molecular weight of 350 Da stabilized the PEI/DNA complex without reducing transfection efficiency [69]. Cracium et al. reported that the PEGylation of PEI reduced the surface charges of the polymer, thus improving its solubility, but also reduced nonspecific ionic interactions between the complex and the target cells [71]. 

### 3.5. FA Modification

Targeted drug delivery systems effectively deliver drugs to the lesion, thereby reducing the damage to normal tissue. FA molecules specifically target cancer cells expressing high levels of FA receptors, which are found on the surface of various types of human cancer cells such as HeLa and KB cells [83]. It is well known that FA-modified NPs have higher specificity and cellular internalization capacity for cancer cells expressing high levels of FA receptors [84,85]. Yang and coworkers conjugated PEI with FA and oleic acid (OA) as a carrier of LOR-2501 for antisense oligonucleotide delivery [72]. Here, OA significantly improved the transfer efficiency of LOR-2501. FA-modified PEI–OA showed a higher level of cellular uptake than PEI and PEI–OA. In our previous work, FA was used to modify PEI through a PEG spacer as a nanoplatform to load the anticancer drug doxorubicin (DOX) for targeted chemotherapy of tumors in vivo (Figure 4a) [57]. As shown in Figure 4b, HeLa cells treated with the FA-targeted PEI/DOX complexes captured more DOX than those treated with the nontargeted PEI/DOX complexes. In addition, under the same conditions, the FA-targeted PEI/DOX complexes were more likely to disrupt the cytoskeleton than the nontargeted PEI/DOX.

### 3.6. HA Modification

HA is the main component of polysaccharide and extracellular matrix with high biodegradability and biocompatibility [86,87]. In addition, it is a broad-spectrum targeting ligand that targets cancer cells overexpressing CD44 receptors [86,87,88,89]. Therefore, HA-targeted NPs are often designed for biomedical applications [50,90,91,92,93,94,95]. Furthermore, HA-modified NPs can effectively prevent plasma protein adsorption and prolong the blood circulation time of NPs [96]. PEI with high positive charges was shielded with HA via electrostatic interaction to deliver DNA effectively and safely into human mesenchymal stem cells (hMSCs) [74]. The HA-shielded PEI/pDNA complexes were easily internalized by the hMSCs and HeLa cells, and the effect was weakened by pretreatment with an anti-CD44 monoclonal antibody. In another study, Liang et al. constructed a self-assembled ternary complex using pDNA, branched PEI, and HA-epigallocatechin gallate (HA-EGCG) conjugates for targeted gene delivery, as shown in Figure 5a [75]. HA not only stabilized the pDNA/PEI complexes through the strong DNA-binding affinity of green tea catechins, but also improved their transport to cells overexpressing CD44 through receptor-mediated endocytosis. The HA-modified pDNA/PEI complexes promoted nuclear transport of pDNA more efficiently in CD44-overexpressed cells than the uncoated complexes (Figure 5b). Similar to FA, the HA molecule can also be used to modify PEI through a PEG spacer for biomedical applications [50]. In addition, HA can be directly chemically conjugated with PEI via 1-ethyl-3-(-3-dimethylaminopropyl) carbodiimide hydrochloride/N-hydroxysuccinimide (EDC/NHS) chemistry for cancer theranostics [92]. 

### 3.7. Protein Modification

The high stability and gene transfection rate of PEI/pDNA complexes require the condensation of pDNA into the nanocomplexes. Histone is one kind of protein that has been used for conjugation with PEI via EDC/NHS chemistry for gene delivery [76]. Histone-modified PEI as a carrier showed low cytotoxicity and could effectively bind and condense pDNA. Katayama et al. found that PEI could be modified with different types of peptides through click chemistry in response to PKCα (Figure 6) [77]. The content and quantity of peptide in PEI/peptide conjugates had significant effects on gene transfection. Because of the negative surface charge of living cells, absorption-mediated endocytosis enables efficient uptake of cationic proteins by cells. Other proteins, such as the RGD peptide, can be modified on the surface of PEI via a PEG bridge for targeted imaging applications [58].

## 4. Synthesis of PEI-Based NPs

NP-based drug delivery systems with high biostability, targeting, and biodegradation have significantly improved clinical efficacy [97,98,99,100]. Compared with traditional drug delivery systems, NP-based drug delivery systems can not only improve the water dispersibility and stability of drugs, but also significantly change the distribution and metabolism of drugs in vivo because of their specific size range (1–100 nm). In addition, the way of drug release can be controlled by appropriate design of delivery vehicles, drug molecule types, and loading modes, so as to achieve the best therapeutic effect. More importantly, in view of the specific receptor expression of cancer cells, NP-based drug delivery systems can be modified with targeted ligands to deliver drugs to specific tumor sites, thereby improving drug bioavailability and reducing toxicity to normal tissues.

PEI plays a crucial role in the construction of multifunctional NPs because of its unique structural features and abundant amino groups. Owing to the presence of hydrophobic cavities in hyperbranched PEI, small molecules, metal ions, and metal oxides can be effectively encapsulated to form stable NPs [25,55,56,58,101,102,103,104,105]. For example, Sun et al. used PEGylated PEI to coat carbon nano-onion clusters (CNOCs) for cancer theranostics [106]. The coating of PEGylated PEI can promote the phagocytosis efficiency of cells to the CNOCs. The CNOCs–PEI–PEG showed a cell uptake rate of 2.13 pg/cell, which was much higher than that of PBS and free CNOCs. The CNOCs–PEI–PEG was used as a photothermal and photoacoustic (PA) imaging agent for cancer theranostics because of its excellent photothermal conversion and cell phagocytosis efficiency.

In addition, the positively charged amino groups on the PEI surface can be bound to organic or inorganic anionic materials using electrostatic interaction. The lone electron of the amino group on the PEI surface can also coordinate with different metal atoms or metal ions to stabilize metal ions, metal oxides, or metal elements. For instance, Liu et al. conjugated PEI to the GO surface via amide bonds, which significantly improved the physiological stability and gene transfection rate of the GO [107]. Sun et al. used linear PEI as a reducing and stabilizing agent to prepare AuNPs in a water bath at 60 °C, and studied the particle size changes of the AuNPs by regulating the feeding ratio of PEI and gold salt [108]. Wang et al. systematically studied the preparation method of AuNPs based on hyperbranched PEI [109]. Note et al. further studied the influencing factors of the size of AuNPs based on hyperbranched PEI, and found that AuNPs with a particle size of less than 10 nm could be obtained in either the aqueous or microemulsion phase when heated to 100 °C [110]. The addition of strong reducing agents, such as sodium borohydride, resulted in preparation of AuNPs with diameters ranging from 2–5 nm. In our previous work, we used PEGylated PEI to entrap and stabilize AuNPs (PP–AuNPs, as shown in Figure 7) or metal ions (gadolinium and technetium ions) for in vivo CT or CT/MR (or SPECT/CT) dual-mode imaging of mice [25,55,56,58]. PEI also can stabilize iron oxide NPs and silver NPs for biomedical applications [111,112,113,114,115,116,117,118,119].

## 5. PEI-Based Drug Delivery Systems

Since the FDA first approved liposomal amphotericin B as a delivery system in 1990, various delivery systems have been developed for the treatment of different diseases [120]. The development of nanotechnology provides more options for the design of drug delivery systems. 

As a cationic polymer, PEI can coat or conjugate drug agents, and numerous amino groups on its surface can be modified with various functional modifications [121]. For example, targeting agents, such as FA [122,123], HA [124,125], lactic acid [126,127], transactivating protein [128], and antibodies [129,130], modify PEI to target specific cancer cells; fluorescent labeling molecules, such as fluorescein isothiocyanate (FI), modify PEI for cell marking [57]; and biocompatible agents, such as PEG and oligosaccharides, can improve the biocompatibility of PEI [56,131,132,133,134]. The internal cavity of hyperbranched PEI and the large number of amino groups on the surface can be readily constructed as a nanoplatform which can effectively stabilize or entrap small biological molecules (e.g., DNA, siRNA, drugs) or metal ions. The unique physicochemical properties and low price of PEI promote its wide application in biomedicine. 

### 5.1. PEI-Based Drug Delivery Systems for Cancer Treatment

PEI is a class of large-molecular-weight polymers, among which hyperbranched PEI has a hydrophobic cavity, dendritic three-dimensional structure, and plentiful positively charged amino groups on the surface, which provide the conditions for further chemical modifications [21,135,136,137]. PEI is an effective drug carrier for cancer treatments such as chemotherapy and gene therapy because of its unique structure, commercial availability, and low price. Table 2 provides a detailed summary of PEI-based drug delivery systems for cancer therapy.

#### 5.1.1. Chemotherapy

Multidrug resistance (MDR) is the most common cause of tumor chemotherapy failure, and low drug delivery efficiency is an important cause of MDR. Because of its unique structure, PEI can effectively coat or conjugate anticancer drug molecules [138]. In addition, amino groups on the PEI surface can be functionalized to achieve targeted drug delivery, which further improves the efficiency of drug delivery [139]. Forrest et al. used PEG-modified PEI to stabilize superparamagnetic iron oxide NPs, and then linked DOX with a pH-sensitive hydrazone bond to inhibit the MDR effect [140]. In addition, they studied the effect of different pH on the release of DOX. It was found that DOX release was greater under acidic conditions (pH 4–5), and a DOX nanocomposite system (NP–DOX) had a more sustained drug-release function than free DOX. As shown in Figure 8, for drug-sensitive C6 cells, free DOX was mainly concentrated in the nucleus, whereas NP–DOX was mainly concentrated around the nucleus. For drug-resistant C6-ARD cells, the fluorescence signal of free DOX was not obvious, whereas the fluorescence signal of DOX could still be clearly seen in the nucleus and cytoplasm of drug-resistant cells treated with NP–DOX. This indicated that NP–DOX had a good ability to inhibit the MDR effect. Therefore, PEI-based NP–DOX can effectively enter the cytoplasm and nucleus, and has better antitumor efficacy than free DOX. Huang et al. synthesized star-block copolymer PEI-g-(PLG-b-PEG) with hyperbranched PEI as the core, poly(l-glutamic acid) (PLG) as the inner shell, and PEG as the outer shell [141]. This copolymer was used as a carrier to coat DOX via electrostatic adsorption. It was found that DOX/PEI complexes could continuously release DOX under a certain pH condition, and the cumulative released amount of DOX increased as the pH decreased. In another study, Tsai et al. prepared PLGA NPs loaded with DOX via electrostatic interaction [142]. Then, cationic PEI and anionic polyacrylic acid (PAA) were alternately deposited on the surface of the PLGA/DOX complexes. Because of the proton sponge effect of PEI, the modification of PEI improved the cellular uptake efficiency and endosomal/lysosomal escape effect of the complexes. In addition, PAA modification resulted in pH-dependent drug-release properties of the complexes. In our previous work, we used targeted molecules, such as HA and FA, to modify PEI to coat DOX for targeted drug delivery [50,57]. 

Recently, the codelivery of nucleic acids and drugs by PEI-mediated drug delivery systems has provided a new therapeutic strategy for cancer treatment with higher antitumor activity. For example, twin-arginine translocation (TAT) protein (or FA)-modified PEG-PEI was conjugated with the anticancer drug DOX through a hydrazone bond and loaded with nucleic acids to construct a dual drug-treatment system for combined chemotherapy and gene therapy in order to destroy cancer cells more effectively [128,143]. 

#### 5.1.2. Gene Therapy

Gene therapy involves the alteration of genes inside cells in order to treat diseases or medical disorders. Vectors for gene therapy need to have not only good biocompatibility and stability, but also a strong gene aggregation effect.

As a common nonviral vector for gene therapy, PEI can condense into a complex with pDNA and escape in vivo through the proton sponge effect, thus effectively improving gene transfection efficiency [144]. However, high cytotoxicity limits the application of PEI in gene therapy. To overcome this issue, Wang et al. modified PEI with lithocholic acid (LCA) and HA to reduce its cytotoxicity and used it as a pDNA vector for gene therapy [145]. Here, LCA stabilizes the pDNA/PEI complexes, and HA, a naturally occurring anionic polysaccharide, shields the PEI from additional cationic charges to reduce the cytotoxicity and prevent the pDNA/PEI complexes from binding to serum proteins [146]. In another work, cysteamine-modified AuNPs/siRNA/PEI/HA complexes were designed using a layer-by-layer method for target-specific intracellular delivery of siRNA [147]. The complexes had no obvious cytotoxicity, and their gene-silencing efficiency was very high: up to 80% in the presence of 50 vol % serum. Furthermore, the complexes reduced the level of ApoB mRNA by about 20% in a dose-dependent manner.

PEG is a biocompatible polymer that has often been conjugated onto the surface of PEI to reduce its cytotoxicity. For instance, Cao et al. reported on PP–AuNPs with different Au atom/PEI molar ratios for pDNA gene transfection [51]. It was found that PP–AuNPs had no significant effect on the gene transfection of PEI, and the cytotoxicity of PP–AuNPs was lower than that of PEI alone. Other agents such as negatively charged agents can also be used to modify PEI to improve its biocompatibility. Zhang and colleagues modified the PEI/DNA complex with negatively charged sodium alginate (Alg) for gene delivery (Figure 9a) [148]. By introducing Ca^2+^ ions to neutralize the carboxyl groups on the surface of Alg, the biocompatibility and stability of Alg were effectively improved, and the obtained Ca^2+^/Alg–PEI–DNA was an effective gene delivery system. The long circulation time of Ca^2+^/Alg–PEI–DNA complexes in the blood enhanced the permeability and retention (EPR) effect of the complexes which improved their accumulation in tumor sites (Figure 9b).

**Table 2 jfb-14-00012-t002:** PEI-based drug delivery system for cancer treatment.

Therapeutic Modalities	Therapeutic Agents	Cell Line Models	In Vivo Models	Ref.
Chemotherapy	DOX	HeLa	HeLa	[57]
	MTX	HCT 116	/	[138]
	PTX	HepG2	/	[139]
	DOX	C6	/	[140]
	DOX	HeLa	HeLa	[143]
	DOX	4T1, HepG2	/	[149]
	DOX	A549	/	[142]
	DOX, siRNA	MDA-MB-231, HeLa, EAT	EAT	[150]
	DOX	SKBR3	SKBR3	[151]
Gene therapy	pDNA	HeLa, 16HBE14o−, HepG2	/	[144]
	pDNA	Huh7	Huh7	[145]
	DNA	NIH/3T3	/	[45]
	pDNA	HeLa	/	[51]
	DNA	HeLa, CT26	CT26	[148]
	mRNA	B16-OVA	B16-OVA	[152]
Other therapies	RNase A	MDA-MB-231	/	[153]
	Oxidized mesoporous carbon nanospheres, pDNA	MCF-7	MCF-7	[154]
	CAT-Ce6	T24	T24	[155]
	GO, DTX, anti-miRNA21	MDA-MB-231	/	[156]
	CuS, DTX, CpG	4T1	4T1	[157]
	pDNA, 9B9 mAb	SMMC-7721	SMMC-7721	[158]

#### 5.1.3. Other Therapies

PEI-based drug delivery systems are also used for other therapies, such as photothermal therapy (PTT), photodynamic therapy (PDT), immunotherapy, and combination therapy [159,160,161,162,163,164]. For example, Huang et al. reported that PEI-coated oxidized mesoporous carbon nanospheres were designed for combined PTT and gene therapy of tumors [154]. In another study, Li et al. assembled fluorinated PEI (F–PEI) and chlorin e6 (Ce6)-conjugated catalase (CAT–Ce6) into an NP for PDT of orthotopic bladder tumors postintravesical instillation [155]. The designed NPs showed significant transmembrane, transmucosal, and intratumoral penetration compared with CAT–Ce6 alone or nonfluorinated CAT–Ce6/PEI NPs. Because CAT–Ce6/F-PEI NPs penetrate bladder tumors to decompose endogenous H_2_O_2_, they can effectively relieve tumor hypoxia. Therefore, compared with hematoporphyrin, intravesical infusion of CAT–Ce6/F–PEI NPs can significantly improve the photodynamic treatment effect and reduce systemic toxicity of orthotopic bladder tumors. In another study, oxidized mesoporous carbon nanospheres were used as photothermal agents with strong NIR absorption. Here, PEI was used to coat the nanospheres and combined with pDNA for combined gene therapy and PTT. Additionally, the nanosphere-based photothermal effect enhanced gene release, thus improving gene therapy. Yang et al. used PEI-modified GO and loaded it with DTX and anti-miRNA21 for chemo-gene-photothermal therapy of triple-negative breast cancer (TNBC) [156]. The nanocomposites showed strong stability, high drug loading efficiency, and excellent nucleic acid absorption capacity. More importantly, the synergistic therapy significantly inhibited the growth and migration of TNBC cells. PEI-coated mesoporous copper sulfide loaded with docetaxel and immunoadjuvant CpG was used for targeted synergistic phototherapy and immunotherapy [157]. The nanocomplexes showed a good PDT effect and photothermal conversion ability under 650 nm and 808 nm irradiation, respectively. In addition, the nanocomplexes significantly inhibited tumor growth without obvious side effects. A low dose of DTX loaded in a nanocomplex can promote cytotoxic T lymphocyte (CTL) infiltration, enhance the efficacy of anti-PD-L1 antibody, inhibit myeloid derived suppressor cells (MDSCs), and polarize MDSCs to M1 phenotype, thus enhancing the antitumor efficacy in vivo.

### 5.2. PEI-Based Drug Delivery System for Cancer Imaging

Contrast agents are widely used in molecular imaging to enhance imaging resolution. Owing to its unique physicochemical structure, PEI can effectively stabilize or encapsulate various agents for cancer imaging applications. A variety of imaging contrast agents can be constructed based on PEI including computed tomography (CT), magnetic resonance (MR), and single-photon emission CT (SPECT) imaging [59,165]. This section summarizes the progress of research concerning the use of PEI to construct multifunctional nanosystems as contrast agents for single-modal and multimodal molecular imaging. Table 3 summarizes PEI-based imaging or imaging-guided cancer therapies.

#### 5.2.1. CT Imaging

CT imaging, as a well-established diagnostic imaging technology, has not only strong penetration, high density, and spatial resolution, but also a very convenient image reconstruction process [166,167]. Iodine-based small molecule contrast agents (e.g., Omnipaque) are the most-used contrast agents in clinical diagnostic imaging [168]. However, these iodine-based small molecules have disadvantages such as nephrotoxicity, a short imaging time, and nonspecificity [169,170,171]. With the development of nanotechnology, a large number of NP-based contrast agents such as AuNPs [24,172], bismuth sulfide NPs [173,174], tungsten sulfide nanosheets [175], copper sulfide NPs [176], and ytterbium-based NPs [166] have been designed to overcome these defects. 

PEI-entrapped AuNPs can significantly improve the stability of AuNPs and can be used as an effective CT contrast agent with better X-ray attenuation performance and longer blood circulation time. In our previous work, we used PP–AuNPs for blood pool and tumor CT imaging applications [24]. In this work, PP–AuNPs were synthesized using partially PEGylated PEI as a template, followed by acetylation of the remaining surface amines of PEI. The formed PP–AuNPs with an average size of 1.9–4.6 nm had excellent water dispersibility, colloidal stability, and biocompatibility. Compared with clinically used iodinated small-molecular contrast agents such as Omnipaque, the PP–AuNPs showed higher X-ray attenuation properties and a longer half-decay time (11.2 h in rats), resulting in an imaging time of up to 75 min, thus enabling enhanced blood pool CT imaging. Similarly, AuNPs can be used as effective contrast agents for CT imaging in tumor models because of the EPR effect. Wang et al. optimized the composition and dosage of PP–AuNPs for blood pool, tumor, and lymph node CT imaging [53]. In another work, FA, as a targeted ligand, was modified with PP–AuNPs for targeted tumor CT imaging [25]. Olifirenko et al. studied the potential applicability of PEI-coated Eu_2_O_3_ (PEI@Eu_2_O_3_) and Dy_2_O_3_ (PEI@Dy_2_O_3_) NPs for enhanced CT imaging [177]. Preliminary cytotoxicity assays on L-929 cells showed that PEI@Eu_2_O_3_ and PEI@Dy_2_O_3_ had no significant toxicity at concentrations below 100 μg/mL. Clinical CT analysis showed that PEI@Eu_2_O_3_ NPs (about 8 HU mM^−1^) exhibited higher X-ray attenuation efficiency than PEI@Dy_2_O_3_ NPs (about 5 HU mM^−1^).

#### 5.2.2. MR Imaging

MR imaging technology is an advanced medical diagnostic imaging technology developed in the 1970s with noninvasion, high spatial resolution, and strong tissue penetration, and has thus been widely used in the detection of various human diseases [178,179]. MR imaging contrast agents are an important part of the technology that can improve imaging contrast and sharpness [180]. Commonly used MR contrast agents are divided into signal-enhancing T_1_-weighted MR contrast agents (e.g., gadolinium agents, manganese dioxide, ultra-small iron oxide NPs, etc.) [181,182,183] and signal-attenuating T_2_-weighted MR contrast agents (e.g., magnetic iron oxide NPs) [184]. 

Studies have found that small-molecule gadolinium agents demonstrated short half-decay time, which greatly limited their applications [185,186]. The abundant amines on the surface of PEI can covalently modify a Gd chelator, so as to effectively chelate gadolinium ions for T_1_-weighted MR imaging. Zhou et al. modified the Gd chelator diethylenetriaminepentaacetic acid (DTPA) on the surface of PEGylated PEI, then chelated Gd ions, and finally acetylated the remaining amines to enhance biocompatibility and prolong circulation time [187]. The prepared PEG-PEI.NHAc-DTPA(Gd) could be used not only for T_1_-weighted MR blood pool imaging, but also for T_1_ MR imaging of tumors, as shown in Figure 10.

Magnetic iron oxide NPs are widely used in molecular imaging, but readily aggregate because of their magnetic properties [59,188,189]. Therefore, surface coating of magnetic iron oxide NPs is required to improve their stability in a physiological environment. PEI is an excellent candidate for surface coating of magnetic iron oxide NPs, providing a hydrophilic surface coating that can effectively enhance the contrast of T_2_-weighted MR imaging [190]. Wang and colleagues constructed an amphiphilic PEI conjugated with indocyanine dye Cy5.5, which was used to coat hydrophobic magnetic iron oxide NPs to form a multimodality nanoprobe for cell imaging [191]. The PEI-coated magnetic iron oxide NPs were effectively internalized into the cytoplasm of MCF-7/Adr, and the T_2_ relaxivity of labeled cells (98.2 s^−1^) was much higher than that of unlabeled cells (2.3 s^−1^).

#### 5.2.3. SPECT Imaging

Nuclear medical imaging uses radionuclide-labeled imaging agents or radiopharmaceuticals, which are introduced into the body of living organisms, thus making it possible to monitor physiological and biochemical processes in real time [192]. Compared with traditional morphological imaging (e.g., B-mode ultrasound, CT, and MR imaging), nuclear medical imaging is functional imaging, which can monitor and reflect metabolic and blood flow changes, specific receptor density, and changes in the activity of organs or tissues in real time [193]. Positron emission tomography (PET) and SPECT are examples of nuclear medicine imaging that receive the γ rays emitted by radionuclide agents [194]. However, the spatial resolution and the accumulation rate of small-molecule radionuclide agents in target tissues are still not sufficiently high [195]. 

To overcome these disadvantages, labeling radionuclides on a PEI-based nanoplatform can effectively enhance their imaging performance. As an example, Zhu et al. constructed multifunctional poly (cyclotriphosphazene-co-PEI) nanospheres (PNSs) labeled with radionuclide ^131^I through 3-(4′-hydroxyphenyl) propionic acid-OSu for SPECT imaging of tumors [196]. The PNSs displayed a high ^131^I label efficiency of up to 76.05 % and a favorable colloidal/radio stability. Furthermore, PNSs effectively accumulated at the tumor site, resulting in higher-contrast SPECT imaging of the tumors. It should be noted that SPECT imaging often needs to be combined with other imaging modalities to improve the diagnosis of tumors, as explained below.

#### 5.2.4. Multimodal Imaging

Each imaging modality has its own inherent advantages and disadvantages. For example, CT imaging has the advantages of low cost, high spatial resolution and short image acquisition time, and can provide high-resolution 3D tomography information. However, CT imaging also has some inherent issues, such as poor soft-tissue resolution, high radioactivity during the detection process, and some nephrotoxicity when contrast agents are used at high concentrations [197]. The advantages of MR imaging are high soft-tissue resolution and sensitivity without damage from ionizing radiation. However, MR imaging has low sensitivity, low spatial resolution, long scanning times, and nephrotoxicity due to the use of gadolinium [180,198]. PET and SPECT imaging can obtain physiological and biochemical information from tumor sites, but struggle to achieve high resolution in terms of anatomical information. Given that each imaging mode has its own shortcomings, a single imaging mode can no longer meet the needs of accurate disease diagnosis. Therefore, combining two or more imaging modes is a developing trend in disease diagnosis.

By organically combining CT and MR imaging elements, multifunctional CT/MR dual-modal imaging contrast agents have been designed that can exploit the advantages of the two imaging modalities and further improve the sensitivity and accuracy of disease diagnosis. For example, Shi et al. used PEGylated PEI to entrap AuNPs and stabilize Fe_3_O_4_ NPs to construct a CT/MR dual-modal imaging contrast agent and successfully applied it to MR and CT imaging of blood pools and organs in vivo [49]. Shi et al. also used PEGylated PEI as a template to load AuNPs and gadolinium oxide (Gd_2_O_3_) NPs for dual-modal CT and MR imaging of tumors [83]. The formed PEI@Au/Gd_2_O_3_ NPs had excellent colloidal stability and cytocompatibility, and displayed high X-ray attenuation efficiency and r_1_ relaxivity, enabling them to be used in dual-modal CT/MR imaging of tumors. In our previous work, PEG was modified by PEI and then linked to a Gd chelator (DOTA), which was used as a template to synthesize AuNPs and chelate Gd ions, and finally the PEI were completely acetylated for dual-modal CT/MT imaging applications [56]. The prepared Gd-PP–AuNPs had a particle size of 4.0 nm and displayed excellent colloidal stability and biocompatibility. Because the imaging elements Au and Gd were in a single nanoplatform, the Gd-PP–AuNPs displayed a good X-ray attenuation coefficient and r_1_ relaxation rate, laying the foundation for in vivo CT and MR imaging. For in vivo CT imaging, only veins can be visualized at low doses, whereas at high doses both veins and arteries can be visualized. For in vivo MR imaging, both arteries and veins can be visualized simultaneously even at low doses, but higher resolution can be obtained at higher doses. We also studied FA ligand-modified Gd-PP–AuNPs for targeted tumor dual-modal CT/MR imaging [54].

The organic combination of functional imaging (PET or SPECT) and structural imaging (CT or MR) can obtain a larger amount of tumor imaging information, which is a developing trend in disease diagnosis. For instance, Zhao et al. selected PEI as a platform to entrap AuNPs and label them with radioactive ^99m^Tc for SPECT/CT imaging in vivo [105]. It was found that the acetylated ^99m^Tc-PP–AuNPs were mainly concentrated in the lungs, liver, and spleen, whereas hydroxylated ^99m^Tc-PP–AuNPs were mainly concentrated in the blood, heart, kidneys, and inferior vena cava. Therefore, it was reasonable to assume that PEI could serve as a versatile nanocarrier to load both AuNPs and ^99m^Tc for dual-modal SPECT/CT imaging of different organs in the body. When further combined with RGD protein, the PEI-based nanosystem can be used for efficient, targeted CT/SPECT dual-modal imaging of different α_v_β_3_-integrin-receptor-overexpressing tumors [58].

PA imaging can effectively image the structure and function of biological tissues, which provides a key method to study the morphological structure, physiological and pathological characteristics, and metabolic function of biological tissues, and is particularly suitable for the early detection and monitoring of cancer [199]. Indocyanine green (ICG) is a near-infrared fluorescent dye that has been approved by the Food and Drug Administration for PA imaging applications. Guo and coworkers reported that fluorinated PEI was modified with lactobionic acid and ICG, and labeled with radionuclide ^99m^Tc for ^19^F-MR/SPECT/PA trimodal imaging of the liver in mice [193]. The nanocomposites were rapidly distributed and eliminated, and the radioactivity was mainly accumulated in the liver. Encapsulation of the ICG in the nanocomposites did not change its optical properties. In addition, the nanocomposites were designed for liver disease diagnosis through the targeted triple imaging of liver cells by lactobionic acid modification.

**Table 3 jfb-14-00012-t003:** PEI-based imaging or imaging-guided cancer therapy.

Imaging Types	Imaging Agents	Cell Line Models	In Vivo Models	Ref.
CT	AuNPs	A549	A549	[53]
	AuNPs	MCF-7	MCF-7	[200]
	AuNPs	HeLa	HeLa	[201,202]
	Bi_2_Se_3_ NPs	A549, U14	U14	[203]
MR	Gd ions	KB	KB	[187]
	Superparamagnetic iron oxide nanocrystals	MCF-7/Adr	/	[191]
	Superparamagnetic iron oxide NPs	Chondrolyte cells	/	[204]
	Ultrasmall iron oxide NPs	4T1	4T1	[104]
	Gd(OH)(3)-doped Fe_3_O_4_ NPs	KB	/	[205]
	Fe_3_O_4_ NPs	HepG2	HepG2	[206]
	Fe_3_O_4_ NPs	U87MG, HeLa	U87MG, HeLa	[90]
SPECT	^131^I	4T1	4T1	[196]
	^99m^Tc	C6	C6	[207]
MR/CT	AuNPs, Gd_2_O_3_	HeLa	HeLa	[208]
	Fe_3_O_4_@Au nanostars	HeLa	HeLa	[92]
	Fe_3_O_4_@Au nanocomposites	KB	/	[49]
	Au-Gd NPs	HeLa	HeLa	[54,209]
MR/PA	Gd/CuS	KB	KB	[210]
SPECT/CT	^99m^Tc, AuNPs	HCC-LM3	HCC-LM3	[58]
	^99m^Tc, AuNPs	SKOV-3	/	[105]
	AuNPs, ^131^I	C6	C6	[121]
MR/CT/PA	Fe_3_O_4_ NPs, Au nanostars	HeLa	HeLa	[211]
MR/SPECT/PA	^19^F,^99m^Tc, ICG	HepG2	HepG2	[193]
CT/MR/upconversion luminescence	Yb^3+^- and Gd^3+^-doped UCNPs	A2780	A2780	[212]

### 5.3. PEI-Based Drug Delivery Systems for Cancer Theranostics

The development of nanotechnology provides new strategies with regard to the combination of therapeutic drugs and imaging agents for imaging-guided cancer therapy, namely cancer theranostics [18,213,214]. As a highly cationic polymer, PEI has the advantages of low cost, easy surface functionalization, stable chemical properties, and high loading of small molecules and NPs, enabling it to be used to construct PEI-based drug delivery systems for cancer theranostics. As an example, Shi et al. used an inverse mini-emulsion method to prepare PEI-based hybrid nanogels for incorporation with ultrasmall iron oxide NPs and the anticancer drug DOX for T_1_ MR imaging-guided chemotherapy of tumors [104]. The nanogels displayed excellent water solubility and colloidal stability, high DOX loading efficiency (51.4%), and a pH-dependent release of the DOX with an accelerated release rate under acidic pH. Compared to free ultrasmall iron oxide NPs, the nanogels showed a much higher r_1_ relaxivity at 2.29 mM^−1^ s^−1^. Additionally, under the guidance of T_1_-weighted MR imaging, the nanogels effectively inhibited tumor growth. HA-modified PEI-stable Fe_3_O_4_@Au core–shell nanostars (NSs) were used for trimodal CT-, MR-, and photothermal- imaging-guided PTT of tumors (Figure 11a) [92]. Here, HA-modified PEI provided the NSs with desirable colloidal stability, biocompatibility, and targeted specificity to cancer cells overexpressing CD44 receptors. With the Fe_3_O_4_ core NPs and Au star shell, the NSs could be used as a contrast agent for efficient MR and CT imaging of tumors in vivo (Figure 11b,c). Furthermore, because of the NIR absorption property, the NSs could also be used as a nanoprobe for thermal imaging (Figure 11d,e) and PTT of tumors (Figure 11f–h).

Laponite (LAP) is a synthetic biodegradable nanoclay with a large specific surface area and cation exchange capacity [215]. Combining LAP with PEI not only can improve the drug loading rate of the complex, but also produce good stability. Zhuang and colleagues created PEI-modified LAP using a polylactic acid-PEG-COOH spacer. The PEI-LAP was used as a nanoplatform to embed AuNPs and load DOX for targeted CT imaging and chemotherapy of tumors [201]. The formed nanocomplexes displayed excellent colloidal stability and a high drug loading efficiency of up to 91.0 ± 1.8%, which significantly inhibited the growth of tumors and reduced the side effects of DOX. Alkoxyphenyl acylsulfonamide (APAS) as a zwitterionic polymer can enhance the cellular uptake of NPs at the pH of tumor microenvironment [216]. Zhu et al. used APAS-linked PEI as a vehicle to entrap AuNPs and labeled it with radioactive ^131^I to enhance dual-modal SPECT/CT imaging-guided radiotherapy of tumors [121]. Because of the charge conversion property of APAS, the AuNPs can change from neutral to positively charged in a weak acid environment, thus promoting cellular uptake. In addition, after ^131^I labeling, the therapeutic agents can enhance SPECT/CT dual mode imaging and radiotherapy of tumors in vivo.

## 6. Outlook and Conclusions

Owing to its unique structure and satisfactory water solubility, PEI has a wide range of applications in biomedical fields, such as drug delivery, medical imaging, and gene therapy. Specifically, PEI effectively coats or covalently binds small drug molecules or nucleic acids for drug and gene delivery, or loads imaging agents for tumor diagnosis, such as AuNPs for CT imaging and magnetic iron oxide NPs for MR imaging. In addition, PEI can be loaded with multiple imaging agents for multimodal imaging. For example, a CT/SPECT dual-modal imaging agent was constructed from PEI-loaded AuNPs combined with radioactive ^99m^Tc [58]. Furthermore, PEI can also simultaneously load drug molecules and imaging agents for cancer theranostics.

Notably, because it contains abundant amines, the surface of PEI is easy to functionalize, for example, by modification with targeted ligands to construct specific targeted nanoplatforms, with fluorescent reagents to achieve the labeling of cells or animal organs, and with some biological proteins or PEG to improve drug loading capacity. For instance, for PEI_25K_ alone, each molecule can only load 50 moles of AuNPs, whereas PEGylated PEI can effectively load 400 moles of AuNPs per PEI [24,53].

Since the first successful PEI-mediated oligonucleotide transfer conducted by the group of Jean-Paul Behr, PEI has been derivatized to improve the physicochemical and biological properties of polyplexes [217]. Several PEI transfection agents have been made commercially available, including ExGen500 and jetPEI [218]. Meleshko et al. complexed pDNA with linear PEI at a low molecular weight (8 kDa) for vaccine delivery [219]. This is the first application of PEI as a vector for an idiotypic DNA vaccine in human phase I clinical trials to have been approved by the regional regulatory authorities of the State Committee on Science and Technology of the Republic of Belarus. Although PEI is probably the most promising second-generation non-viral vector, several critical issues need to be addressed before its clinical translation for cancer theranostics. First, PEI itself has obvious cytotoxicity, and various surface modification methods can be used to improve its biocompatibility. However, there is still a lack of systematic research on how to select appropriate surface-modification methods according to the specific research purposes. Second, the type and molecular weight of PEI seriously affect the loading efficiency of drugs, but the relationship among them is still unclear. Third, although various types of targeting agents have been developed, their drug delivery efficiency is still very low (less than 5%). The delivery efficiency of PEI-based drug delivery systems should be improved for cancer theranostics applications. Fourth, current research on PEI-based drug delivery systems is mainly focused on cell or subcutaneous tumor models, and there is a lack of exploration of their applications to orthotopic or human-excision orthotopic tumors. Lastly, PEI-based drug delivery systems have unnoticeable short-term toxicity at the animal level through appropriate surface modifications, but their long-term biosafety and biodegradability should be fully investigated. Designs of PEI-based drug delivery systems that are biodegradable or reduced in size within the renal filtration threshold for rapid renal clearance are encouraged.

## Figures and Tables

**Figure 1 jfb-14-00012-f001:**
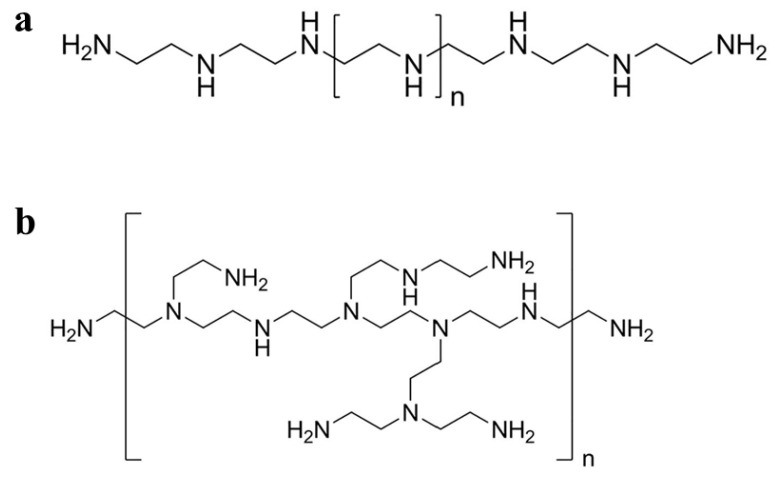
Schematic diagram of the chemical structures of (**a**) linear and (**b**) branched PEI.

**Figure 2 jfb-14-00012-f002:**
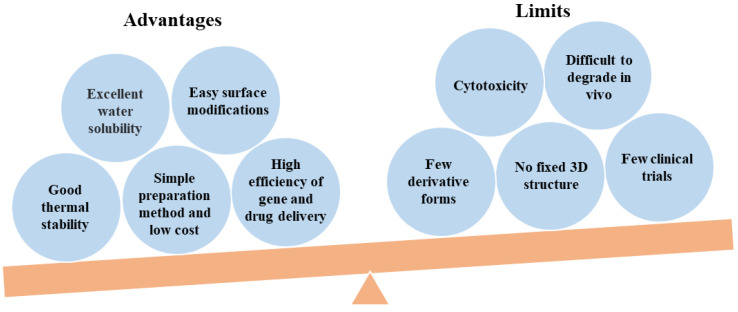
A balanced picture of PEI studies including advantages and limits.

**Figure 3 jfb-14-00012-f003:**
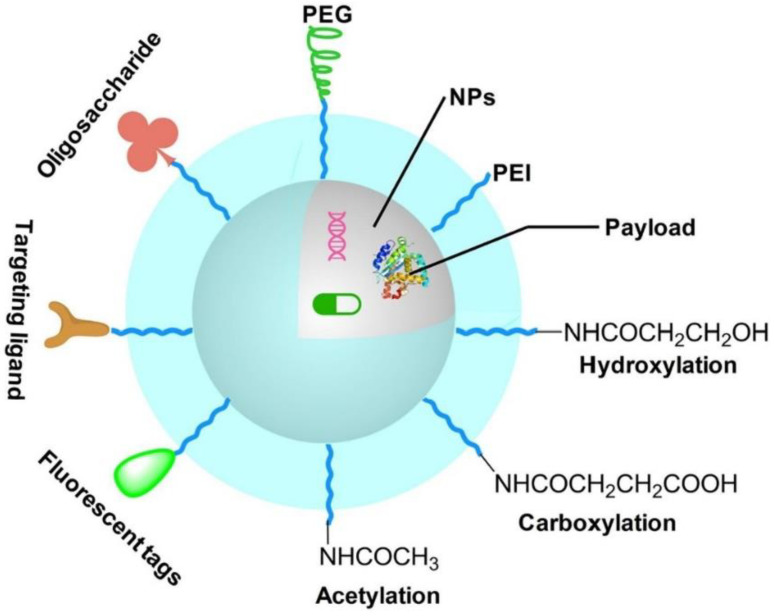
Surface modifications of PEI for biomedical applications. Reproduced with permission of [59]. Copyright 2022, Elsevier Ltd.

**Figure 4 jfb-14-00012-f004:**
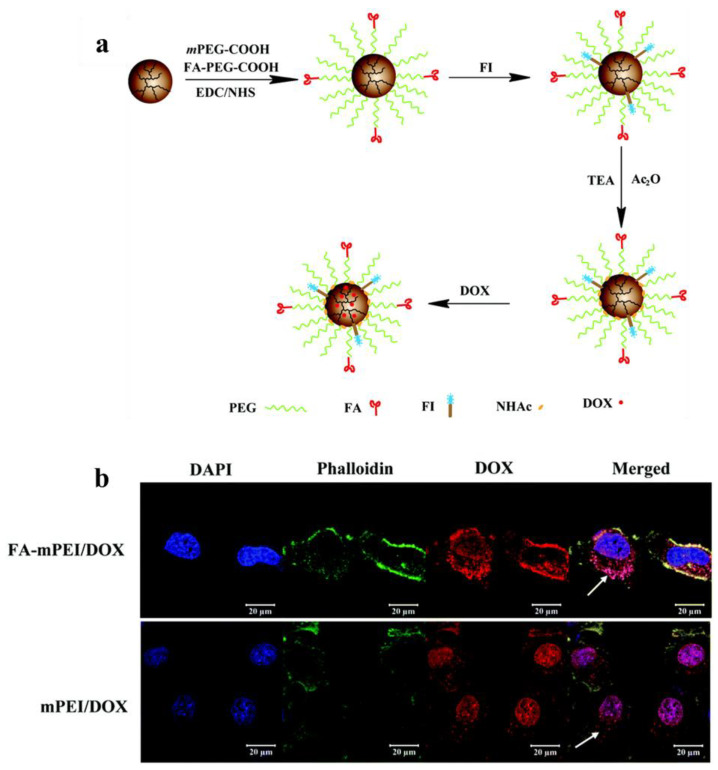
(**a**) Schematic illustration of the preparation of FA-targeted PEI/DOX complexes. (**b**) Confocal microscopic images of HeLa cells treated with FA-targeted or nontargeted PEI/DOX at a DOX concentration of 10 μg/mL (white arrows indicate drugs that were taken up by cells). Reproduced with permission of [57]. Copyright 2017, the Royal Society of Chemistry.

**Figure 5 jfb-14-00012-f005:**
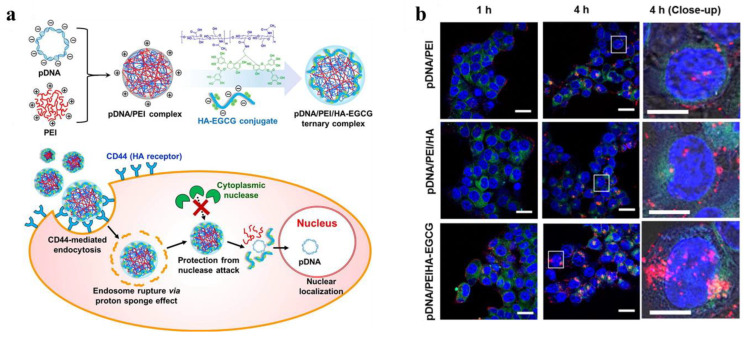
(**a**) Schematic showing the synthesis and mechanism of self-assembled complexes of pDNA/PEI/HA-EGCG for gene delivery. (**b**) Confocal microscope images of HCT-116 cells transfected with different complexes for 1 h and 4 h. The red, blue, and green fluorescence regions show the distribution of Cy5-labeled pDNA, nuclei, and endolysosomal compartments in the cells, respectively. Reproduced with permission of [75]. Copyright 2016, Elsevier B.V.

**Figure 6 jfb-14-00012-f006:**
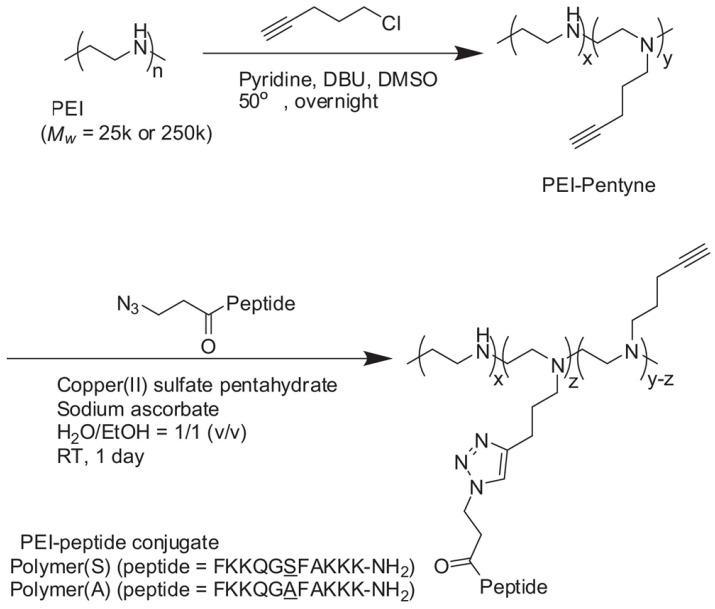
Synthetic scheme for PEI–peptide conjugates. Reproduced with permission of [77]. Copyright 2014, Elsevier B.V.

**Figure 7 jfb-14-00012-f007:**
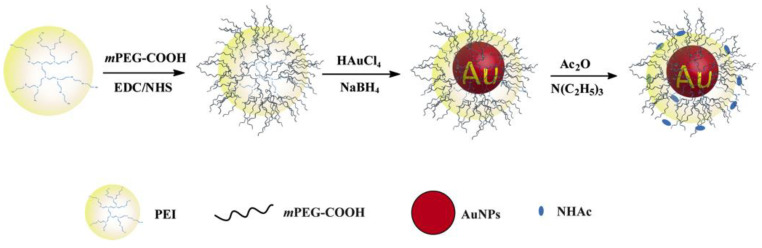
Schematic illustration of the synthesis of PP–AuNPs.

**Figure 8 jfb-14-00012-f008:**
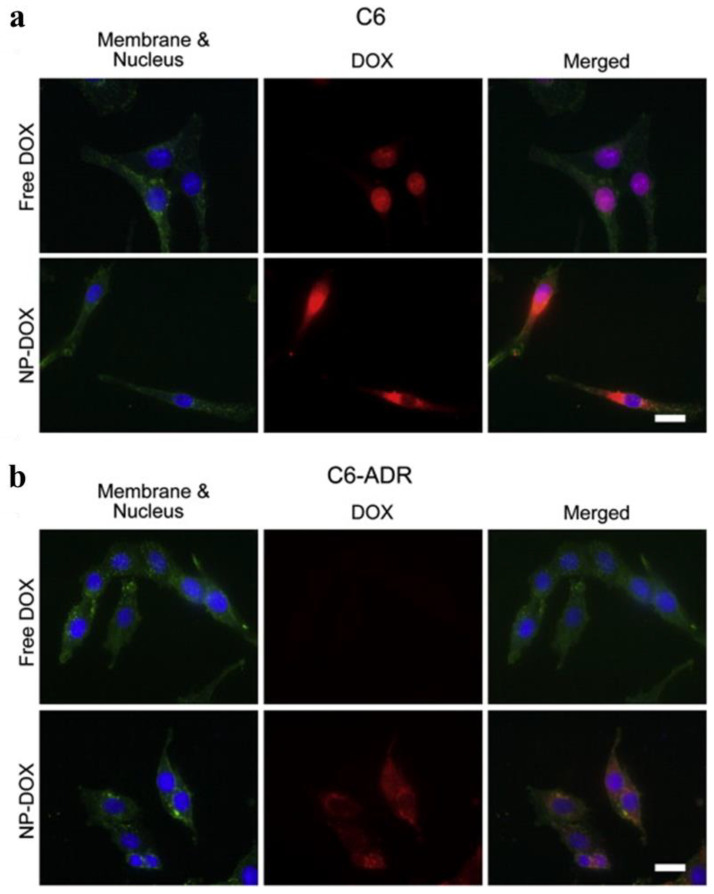
Fluorescence images of (**a**) DOX-sensitive C6 cells and (**b**) drug-resistant C6-ARD cells treated with 1 μg/mL of DOX and the same amount of DOX in NP–DOX for 4 h. Scare bar = 20 μm. Reproduced with permission of [140]. Copyright 2011, Elsevier B.V.

**Figure 9 jfb-14-00012-f009:**
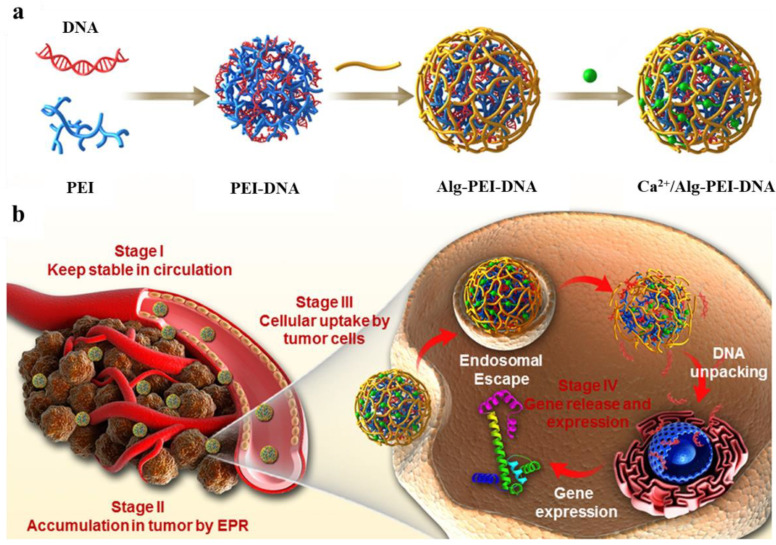
Schematic illustration of (**a**) preparation of Ca^2+^/Alg–PEI–DNA complexes and (**b**) the transportation process of Ca^2+^/Alg-PEI-DNA complexes in vivo. Reproduced with permission of [148]. Copyright 2018, Elsevier Ltd.

**Figure 10 jfb-14-00012-f010:**
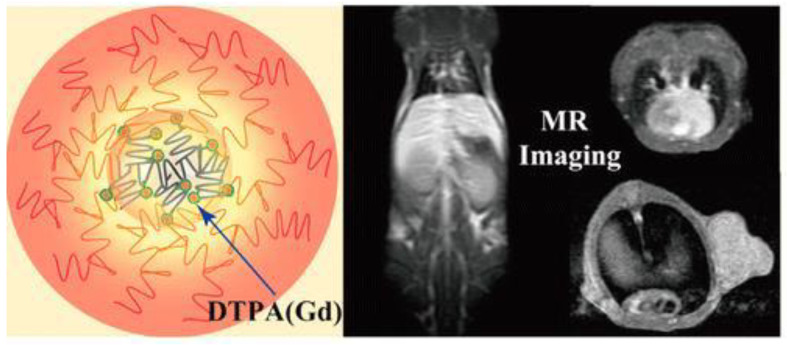
Structure illustration of PEG-PEI.NHAc-DTPA(Gd) and T_1_-weighted MR imaging in vivo. Reproduced with permission of [187]. Copyright 2014, American Chemical Society.

**Figure 11 jfb-14-00012-f011:**
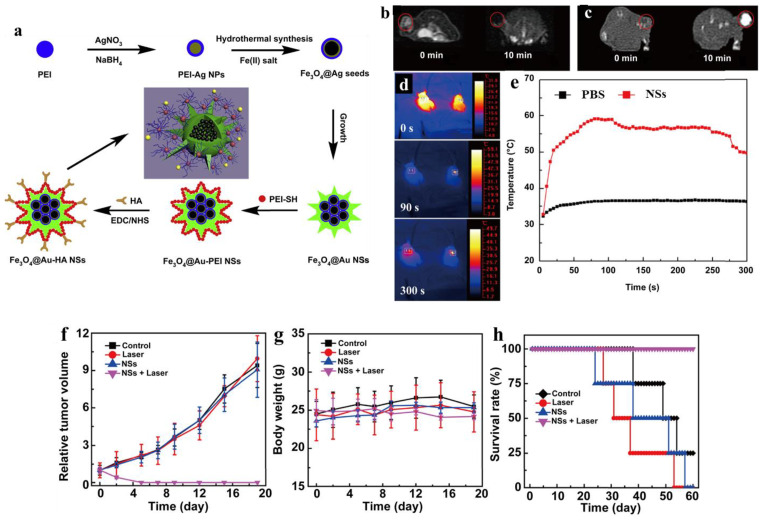
(**a**) Schematic illustration of the preparation of Fe_3_O_4_@Au-PEI-HA NSs. (**b**) T_2_-weighted MR, (**c**) CT, and (**d**) photothermal images of the tumors after intratumoral injection of the NSs. (**e**) The temperature curves of PBS and the NSs as a function of the laser irradiation time. The relative (**f**) tumor volume, (**g**) body weight, and (**h**) survival rate of tumor-bearing mice after different treatments. Reproduced with permission of [92]. Copyright 2014, Elsevier Ltd.

**Table 1 jfb-14-00012-t001:** Summary of PEI modifications carried out in recent years.

Modification Types	Aims	Ref.
Carboxylation modification	Gene delivery, absorption of heavy metals in sewage.	[60,61,62,63]
Acetylation modification	Gene delivery efficiency improvement, cytotoxicity reduction.	[63,64,65]
Hydroxylation modification	Biocompatibility enhancement, gene delivery, transformation improvement of NPs.	[66,67,68]
PEG modification	Stability and transfection efficiency improvement.	[69,70,71]
FA modification	Tumor-targeted delivery.	[72,73]
HA modification	Tumor-targeted gene delivery, stability improvement.	[74,75]
Protein modification	Gene delivery, protein transduction.	[76,77,78]
FI modification	Fluorescence imaging.	[57]

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
