# Peer review of "Polyethyleneimine-Based Drug Delivery Systems for Cancer Theranostics"

_jfb, 2022, doi:10.3390/jfb14010012_

Round 1

Reviewer 1 Report

The paper is well written. It covers the up-to-date research. The reviewer recommends the paper for publication.

Author Response

We thank the reviewer for his/her support of our manuscript.

Reviewer 2 Report

Dear authors,

The manuscript is of interest and well organized. The bibliography is diverse, but you can include more recent data.

This review is a literature review and not a systematic review. Please revise this, for example, line 52.

Please carefully revise the sentences because some of them are too long. Example: abstract lines 14-16.

Please verify if you put the abbreviations in complete form when it appears first, eg, line 110, PLGA, among others.

Line 203: is the cell membrane of the cells positive? Maybe there is misleading info here. Carefully check this.

I recommend using different words than discovered.

Please avoid repetition of some ideas like:

“PEI as a highly cationic polymer has the advantages of low cost, easy surface functionalization, stable chemical properties, and high loading of small molecules and NPs, which can be used to construct PEI-based drug delivery systems for cancer theranostics.”

[lines 381-383, 534-536,…]

What is a low-cost polymer? What about the scaling up?

Carbon nanotubes (line 38) are nanomaterials, particularly, nano-objects, and they are not nanoparticles according to the ICH guideline. Please check and revise this info.

I recommend including a topic regarding the main toxic effects of PEI and differentiating between branched and linear forms, eg, cytotoxicity and immunogenicity. After that, add the use of other decorations to overcome these limitations. Is the polymer biodegradable in the human body or in the environment? Could you please clarify it from a sustainable-by-design perspective?

Correct PH to pH (line 303, …)

Line 243: revise “addition of additional”. It seems to be redundant.

Revise lines 317-318.

Consider revising lines 333-334 and similar

“PEG with excellent biocompatibility has often been conjugated onto the surface of PEI to reduce the cytotoxicity of PEI” like this: PEG is a biocompatible polymer that has often been conjugated onto the surface of PEI to reduce its cytotoxicity”.

Table 2 and 3: What “/” means?. Correct “tumor modes”. Add the cell lines/in vivo models used and indicate them.

How far are we from the clinical translation?

Author Response

Comment 1:

The manuscript is of interest and well organized. The bibliography is diverse, but you can include more recent data.

Author reply: We thank the reviewer for his/her great comments. According to the reviewer’s suggestion, we have updated the bibliography and cited more recent literatures in the revised manuscript.

Comment 2:

This review is a literature review and not a systematic review. Please revise this, for example, line 52.

Author reply: According to the reviewer’s suggestion, we have revised them in the revised manuscript.

Comment 3:

Please carefully revise the sentences because some of them are too long. Example: abstract lines 14-16.

Author reply: According to the reviewer’s suggestion, we have revised the sentences in abstract lines 13-15 in the revised manuscript. See below:

“As an effective vehicle for drug delivery systems, polyethyleneimine (PEI) has been extensively studied due to its rich surface amines and excellent water solubility.”

Comment 4:

Please verify if you put the abbreviations in complete form when it appears first, e.g., line 110, PLGA, among others.

Author reply: We have added the abbreviations in complete form when they appear first in the revised manuscript.

Comment 5:  

Line 203: is the cell membrane of the cells positive? Maybe there is misleading info here. Carefully check this.

Author reply: We are sorry for this mistake and have corrected it in the revised manuscript. See below:

Lines 219-220:

Because of the negative surface charge of living cells, absorption-mediated endocytosis enables efficient uptake of cationic proteins by cells

Comment 6:

I recommend using different words than discovered.

Please avoid repetition of some ideas like:

“PEI as a highly cationic polymer has the advantages of low cost, easy surface functionalization, stable chemical properties, and high loading of small molecules and NPs, which can be used to construct PEI-based drug delivery systems for cancer theranostics.”

[lines 381-383, 534-536…]

Author reply: We thank the reviewer for his/her great comments. To address the review’s concerns, we have used other words, such as reported or found, to replace discovered in the revised manuscript. In addition, we revised the sentences in lines 398-400 in the revised manuscript using different words from lines 554-557. See also below:

“Contrast agents are widely used in molecular imaging to enhance imaging resolution. Owing to its unique physicochemical structure, PEI can effectively stabilize or encapsulate various agents for cancer imaging applications.”

Comment 7:

What is a low-cost polymer? What about the scaling up?

Author reply: To address the review’s concerns, we have added description in lines 62-65 in the revised manuscript. See blow:

“PEI can be easily prepared using an AB-type monomer via a simple one-step reaction [24]. In addition, PEI is a low-cost polymer with a price of around 1000 RMB per 100 mL, which is much lower than that of dendrimers with the same molecular weight.”

Comment 8:

Carbon nanotubes (line 38) are nanomaterials, particularly, nano-objects, and they are not nanoparticles according to the ICH guideline. Please check and revise this info.

Author reply: We are sorry for this mistake. We have checked and corrected this information in the revised manuscript.

Comment 9:

I recommend including a topic regarding the main toxic effects of PEI and differentiating between branched and linear forms, e.g., cytotoxicity and immunogenicity. After that, add the use of other decorations to overcome these limitations. Is the polymer biodegradable in the human body or in the environment? Could you please clarify it from a sustainable-by-design perspective?

Author reply: According to the reviewer’s suggestion, we have added the main toxic effects of PEI and the difference between branched and linear forms in lines 94-101 in the revised manuscript. See blow:

“Cationic PEI enters cells by adhering to negatively charged transmembrane heparanproteoglycans, which can cause cell damage through membrane destabilization [44]. Additionally, the internalized PEI causes apoptotic cell death by forming pores in the mitochondrial membrane [45,46]. PEI is not well degraded in organisms, and its cytotoxicity is closely related to the molecular weight and level of branched degree of PEI [47]. Branched PEI with higher molecular weight has higher cytotoxicity. The surface amines of PEI can be shielded by simple modifications, thus significantly improving the biocompatibility of PEI [21].”

Comment 10:

Correct PH to pH (line 303, …)

Author reply: We have corrected it in the revised manuscript.

Comment 11:

Line 243: revise “addition of additional”. It seems to be redundant.

Author reply: We have corrected it in the revised manuscript.

Comment 12:

Revise lines 317-318.

Author reply: According to the reviewer’s suggestion, we have revised the sentence in lines 334-335 in the revised manuscript. See blow:

“Gene therapy involves the alteration of genes inside cells in order to treat disease or medical disorders.”

Comment 13:

Consider revising lines 333-334 and similar

“PEG with excellent biocompatibility has often been conjugated onto the surface of PEI to reduce the cytotoxicity of PEI” like this: PEG is a biocompatible polymer that has often been conjugated onto the surface of PEI to reduce its cytotoxicity”.

Author reply: We thank the reviewer for his/her great comments. According to the reviewer’s suggestion, we have revised the sentence in lines 350-351 in the revised manuscript. See below:

“PEG is a biocompatible polymer that has often been conjugated onto the surface of PEI to reduce its cytotoxicity.”

Comment 14:

Table 2 and 3: What “/” means? Correct “tumor modes”. Add the cell lines/in vivo models used and indicate them.

Author reply: “/” means no in vivo tumor model. To address the review’s concerns, we have added cell lines/in vivo tumor models in Tables 2 and 3 in the revised manuscript.

Comment 15:

How far are we from the clinical translation?

Author reply: We thank the reviewer for his/her great comments. To address the review’s concerns, we have added description in lines 606-622 in the revised manuscript. See below:

In general, PEI-based drug delivery systems still have a long way to go before clinical translation. Several critical issues need to be addressed before they can be translated into clinical trials. First, PEI itself has obvious cytotoxicity, and various surface modification methods can be used to improve its biocompatibility. However, there is still a lack of systematic research on how to select appropriate surface modification methods according to specific research purposes. Second, the type and molecular weight of PEI seriously affect the loading efficiency of drugs, but the relationship among them is still unclear. Third, although various types of targeting agents have been developed, their drug delivery efficiency is still very low (less than 5%). The delivery efficiency of PEI-based drug delivery systems should be improved for cancer theranostics applications. Fourth, Current research on PEI-based drug delivery systems is mainly focused on cell or subcutaneous tumor models, and there is a lack of exploration of their applications to orthotopic or human excision orthotopic tumors. Lastly, PEI-based drug delivery systems have unnoticeable short-term toxicity at the animal level through appropriate surface modifications, but their long-term biosafety and biodegradability should be fully investigated. Designs of PEI-based drug delivery systems that are biodegradable or reduced in size within the renal filtration threshold for rapid renal clearance are encouraged.”

Reviewer 3 Report

The authors present a review of PEI characteristics, synthesis methods, and applications for cancer treatment and imaging. The review is well organized and covers an appropriate scope. There are two issues which should be addressed.

1. An impactful review article provides both a detailed background (which the authors have done well) and a critical viewpoint, which is limited in this manuscript. For example, in section 3, when multiple approaches are used for a common goal such as cytotoxicity reduction, the authors should contrast the benefits and challenges of the approaches so that the reader can better understand the technical issues.

2. The review would be more impactful if it contained an additional section on clinical trial results, including both successes and failures. This would provide the reader with a sense of how the research field has progressed out of the laboratory and what further challenges are ahead. 

Author Response

Comments to the Author

The authors present a review of PEI characteristics, synthesis methods, and applications for cancer treatment and imaging. The review is well organized and covers an appropriate scope. There are two issues which should be addressed.

Comment 1:

An impactful review article provides both a detailed background (which the authors have done well) and a critical viewpoint, which is limited in this manuscript. For example, in section 3, when multiple approaches are used for a common goal such as cytotoxicity reduction, the authors should contrast the benefits and challenges of the approaches so that the reader can better understand the technical issues.

Author reply: We thank the reviewer for his/her great comments. To address the review’s concerns, we have added description in line 101-105 in the revised manuscript. See below:

“At present, the surface amines of PEI are mainly shielded by covalent bonds such as carboxylation, acetylation and hydroxylation, or by electrostatic modification of negatively charged proteins. However, currently, there is a lack of systematic research to contrast the benefits and challenges of these approaches for the surface modifications of PEI.

Comment 2:

The review would be more impactful if it contained an additional section on clinical trial results, including both successes and failures. This would provide the reader with a sense of how the research field has progressed out of the laboratory and what further challenges are ahead.

Author reply: We thank the reviewer for his/her great comments. To our knowledge, there is a lack of clinical trial reports on PEI-based drug delivery systems for cancer theranostics. The reason has been explained in lines 606-622. See below:

“In general, PEI-based drug delivery systems still have a long way to go before clinical translation. Several critical issues need to be addressed before they can be translated into clinical trials. First, PEI itself has obvious cytotoxicity, and various surface modification methods can be used to improve its biocompatibility. However, there is still a lack of systematic research on how to select appropriate surface modification methods according to specific research purposes. Second, the type and molecular weight of PEI seriously affect the loading efficiency of drugs, but the relationship among them is still unclear. Third, although various types of targeting agents have been developed, their drug delivery efficiency is still very low (less than 5%). The delivery efficiency of PEI-based drug delivery systems should be improved for cancer theranostics applications. Fourth, Current research on PEI-based drug delivery systems is mainly focused on cell or subcutaneous tumor models, and there is a lack of exploration of their applications to orthotopic or human excision orthotopic tumors. Lastly, PEI-based drug delivery systems have unnoticeable short-term toxicity at the animal level through appropriate surface modifications, but their long-term biosafety and biodegradability should be fully investigated. Designs of PEI-based drug delivery systems that are biodegradable or reduced in size within the renal filtration threshold for rapid renal clearance are encouraged.”

Round 2

Reviewer 2 Report

Dear authors,

Please address some minor recommendations.

I will perform my suggestions using the numeric order of the comments presented in your reply.

Comment 3:

“As an effective vehicle for drug delivery systems, polyethyleneimine (PEI) has been extensively studied due to its rich surface amines and excellent water solubility.”

Maybe you can include the transfection capabilities of PEI, like this:

“As an effective vehicle for drug and gene delivery, polyethyleneimine (PEI) has been extensively studied due to its rich surface amines and excellent water solubility.”

Comment 7

“PEI can be easily prepared using an AB-type monomer via a simple one-step reaction [24]. In addition, PEI is a low-cost polymer with a price of around 1000 RMB per 100 mL, which is much lower than that of dendrimers with the same molecular weight.”

I would like to recommend removing the price and writing something like this:

“PEI can be easily prepared using an AB-type monomer via a simple one-step reaction [24]. In addition, PEI can be considered a low-cost option compared to…”

and adding the appropriate reference that supports this statement.

Comment 14

Please clearly separate the cell line models in a column and the in vivo models in another one. I think it will be of benefit to the readers.

Comment 15

I would like to suggest the addition of references that support the statements present in this section.

Some clinical trials report the use of PEI, did you find some results? Is PEI approved for medicinal purposes by any regulatory agency?

Author Response

Reviewer #1:

Comment 1:

“As an effective vehicle for drug delivery systems, polyethyleneimine (PEI) has been extensively studied due to its rich surface amines and excellent water solubility.”

Maybe you can include the transfection capabilities of PEI, like this:

“As an effective vehicle for drug and gene delivery, polyethyleneimine (PEI) has been extensively studied due to its rich surface amines and excellent water solubility.”

Author reply: We thank the reviewer for his/her great comments. According to the reviewer’s suggestion, we have revised the sentences in lines 13-15 in the revised manuscript. See below:

“As an effective vehicle for drug and gene delivery, polyethyleneimine (PEI) has been extensively studied due to its rich surface amines and excellent water solubility.”

Comment 2:

“PEI can be easily prepared using an AB-type monomer via a simple one-step reaction [24]. In addition, PEI is a low-cost polymer with a price of around 1000 RMB per 100 mL, which is much lower than that of dendrimers with the same molecular weight.”

I would like to recommend removing the price and writing something like this:

“PEI can be easily prepared using an AB-type monomer via a simple one-step reaction [24]. In addition, PEI can be considered a low-cost option compared to…”

and adding the appropriate reference that supports this statement.

Author reply: According to the reviewer’s suggestion, we have revised the sentences in lines 62-65 and added relevant references in the revised manuscript. See below:

“PEI can be easily prepared using an AB-type monomer via a simple one-step reaction [24]. In addition, PEI can be considered a low-cost option compared to dendrimers with the same molecular weight [25].”

Comment 3:

Please clearly separate the cell line models in a column and the in vivo models in another one. I think it will be of benefit to the readers.

Author reply: According to the reviewer’s suggestion, we have separated the cell line models from the in vivo models in Tables 2 and 3 in the revised manuscript.

Comment 4:

I would like to suggest the addition of references that support the statements present in this section.

Some clinical trials report the use of PEI, did you find some results? Is PEI approved for medicinal purposes by any regulatory agency?

Author reply: We thank the reviewer for his/her great comments. To address the review’s concerns, we have added some relevant clinical trials about the use of PEI in lines 606-613 in the revised manuscript. See blow:

“Since the first successful PEI-mediated oligonucleotide transfer conducted by the group of Jean-Paul Behr, PEI has been derivatized to improve the physicochemical and biological properties of polyplexes [217]. Several PEI transfection agents have been made commercially available, including ExGen500 and jetPEI [218]. Meleshko et al. complexed pDNA with linear PEI at a low molecular weight (8 kDa) for vaccine delivery [219]. This is the first application of PEI as a vector for idiotypic DNA vaccine in human phase I clinical trials, which has been approved by the regional regulatory authorities of State Committee on Science and Technology of the Republic of Belarus.”

Reviewer 3 Report

The authors provided a heavily edited revision to the original manuscript. The revision reads more clearly and addresses most of the reviewer comments. The manuscript still seems optimistic in its presentation (until discussing clinical prospects), with an emphasis on what can be done and less information about limits and tradeoffs. Such information would be especially helpful to the reader when trying to sort through the large volume of prior studies. The authors are encouraged to present a balanced picture of the PEI studies. For example, in Table 1, add a column next to 'Aims' which lists major challenges or limits. This kind of balancing would be helpful in all subsequent sections as well. 

Author Response

Reviewer #2:

Comment 1:

The authors provided a heavily edited revision to the original manuscript. The revision reads more clearly and addresses most of the reviewer comments. The manuscript still seems optimistic in its presentation (until discussing clinical prospects), with an emphasis on what can be done and less information about limits and tradeoffs. Such information would be especially helpful to the reader when trying to sort through the large volume of prior studies. The authors are encouraged to present a balanced picture of the PEI studies. For example, in Table 1, add a column next to 'Aims' which lists major challenges or limits. This kind of balancing would be helpful in all subsequent sections as well.

Author reply: We thank the reviewer for his/her great comments. To address the review’s concerns, we have presented a balanced picture of the PEI studies including advantages and limits in Figure 2 in the revised manuscript.
